# Cardiovascular Anomalies among 1005 Fetuses Referred to Invasive Prenatal Testing—A Comprehensive Cohort Study of Associated Chromosomal Aberrations

**DOI:** 10.3390/ijerph191610019

**Published:** 2022-08-14

**Authors:** Anna Wójtowicz, Anna Madetko-Talowska, Wojciech Wójtowicz, Katarzyna Szewczyk, Hubert Huras, Mirosław Bik-Multanowski

**Affiliations:** 1Department of Obstetrics & Perinatology, Jagiellonian University Medical College, 31-501 Kraków, Poland; 2Department of Medical Genetics, Jagiellonian University Medical College, 30-663 Kraków, Poland; 3Information Technology Systems Department, Faculty of Management and Social Communication, Jagiellonian University, 30-348 Kraków, Poland

**Keywords:** congenital heart defect, vascular anomaly, array comparative genomic hybridization, copy number variants, prenatal diagnosis, ultrasound

## Abstract

This retrospective cohort study comprehensively evaluates cardiovascular anomalies (CVAs) and associated extracardiac structural malformations (ECMs) among 1005 fetuses undergoing invasive prenatal testing at a single tertiary Polish center in the context of chromosomal aberrations detected in them by array comparative genomic hybridization (aCGH) and G-band karyotyping. The results of our study show that CVAs are among the most common malformations detected in fetuses undergoing invasive prenatal testing, as they affected 20% of all cases seen in our department. Septal defects predominated among fetuses with numerical aberrations, while conotruncal defects were the most common findings among fetuses with pathogenic copy number variants (CNVs). In 61% of cases, CVAs were associated with ECMs (the diagnosis was confirmed postnatally or in cases of pregnancy termination by means of autopsy). The most common ECMs were anomalies of the face and neck, followed by skeletal defects. In total, pathogenic chromosomal aberrations were found in 47.5% of CVAs cases, including 38.6% with numerical chromosomal aberrations. Pathogenic CNVs accounted for 14.5% of cases with CVAs and normal karyotype. Thus, our study highlights the importance of assessing the anatomy of the fetus, and of the genetic testing (preferably aCGH) that should be offered in all CVA and ECM cases.

## 1. Introduction

Congenital heart defects (CHDs) are the most common life-threatening birth defects and the leading cause of non-infectious neonatal mortality [1], affecting 6.5 per 1000 live-born newborns in European countries [2]. Heart defects may be accompanied by vascular anomalies such as single umbilical artery (SUA), ductus venosus (DV) agenesis, and persistent left superior vena cava (PLSVC), which are also common, occurring with frequencies of 0.2–1.5%, depending on the type of population studied and when the study was performed [3,4,5].

Studies have shown the importance of the prenatal detection of cardiovascular anomalies (CVAs), which allows for multidisciplinary consultation, planning of the time and place of delivery, and the planning of postnatal surgical procedures [6]. Today, in most cases of CHDs, surgical repair or palliation is possible with a good outcome, but CHDs remain the leading cause of birth-defect-related morbidity, as well as infant deaths [7].

An important factor influencing the prognosis and treatment plan is whether the defect is isolated or coexists with other anomalies, especially with chromosomal abnormalities [1,8,9]. Previous studies have shown that extracardiac malformations (ECMs) may be present in up to 60% of live-born fetuses with CHDs and associated genetic syndromes in about 30% of neonates and children with congenital heart defects [10,11]. Genetic causes of CHDs include numerical chromosomal aberrations, such as trisomies 13,18, and 21; Turner syndrome; and structural aberrations such as 22q11 microdeletion [11,12]. Additionally, in the last decade, submicroscopic deletions or duplications, commonly referred to as copy number variants (CNVs) that are detected in chromosome microarray analysis (CMA) have been reported in children with CHDs [13,14] and in about 12% of prenatally diagnosed CHDs [15]. CNVs can have a major impact on neurological development, quality of life, and the life expectancy of children with CHDs [16]. Therefore, the prenatal assessment of the presence of pathogenic CNVs may be crucial for prognostic purposes for the parents.

Array comparative genomic hybridization (aCGH), which is the most commonly used chromosome microarray analysis (CMA) technique, enables the identification of the whole spectrum of CNV abnormalities, from aneuploidy to very small, submicroscopic aberrations (microdeletions and/or microduplications) that are not routinely seen on karyotyping. Classic G-band karyotyping is used to detect chromosomal aberrations at a resolution of approximately 5 Mb, while CMA has a typical resolution of ~50–100 kb for genomic imbalances. In recent years, CMA has been introduced as a first-tier diagnostic tool for the evaluation of neurocognitive disabilities and congenital structural anomalies in children [17,18,19,20], with an additional diagnostic yield of 12–15% of genetic causes in comparison to standard G-band karyotyping. Recently, CMA has also been recommended as a first-tier test for identifying CNVs in the prenatal setting for women undergoing prenatal screening, even in the setting of a normal fetal ultrasound [21,22,23,24,25,26,27].

Therefore, the aim of this study is to assess the frequency and scope of CNVs, determine potential causes of CHDs among fetuses undergoing invasive prenatal testing, and comprehensively analyze this population regarding associated congenital defects as well as neonatal outcomes.

## 2. Materials and Methods

This was a retrospective cohort study carried out at one tertiary Polish center for the prenatal diagnosis and management of fetal and neonatal pathology: the Department of Obstetrics and Perinatology in Cracow, Poland. Our center has been conducting screening for aneuploidy financed by the National Health Fund for patients from south-eastern Poland since 2004. This study is a retrospective analysis of the medical records of pregnant women who underwent invasive prenatal testing between 2018 and 2021. In the analyzed period, 12,776 pregnant women were screened, of whom 1005 were subjected to invasive prenatal testing after genetic counseling: chorionic villous sampling (15 cases) or amniocentesis (990 cases).

All pregnant women included in the study were Caucasian. Informed consent was obtained from all the participants of this study.

An ultrasound examination was performed in each case before and after invasive testing. First-trimester screening and second-trimester screening were carried out following Polish and international guidelines [28,29]. Nuchal translucency (NT) above the 95th percentile was considered abnormal. Additionally, the first-trimester ultrasound assessment included blood flow through the tricuspid valve (TV) and DV, presence/absence of nasal bone (NB), and the anatomy of the fetuses. During first-trimester screening, in each case, serum markers such as free β-human chorionic gonadotrophin (free β-hCG) and pregnancy-associated plasma protein A (PAPP-A) were measured according to the guidelines [28,29]. Each patient was also informed about the possibility of non-invasive prenatal genetic testing (NIPT) as a screening method. However, these tests are not reimbursed by the Polish National Health Fund and were performed in only 21 cases. Second-trimester screening was based on ultrasound only.

Ultrasound examinations were carried out using a Voluson E6 (GE Healthcare Medical Systems, Milwaukee, WI, USA) system and included a detailed assessment of cardiac and noncardiac structures according to national guidelines [28] and the International Society of Ultrasound in Obstetrics and Gynecology [29,30]. If CVAs were suspected, echocardiography was performed. Fetal heart examinations were performed using conventional two-dimensional ultrasound and color and pulsed-wave Doppler ultrasound by cardiologists and physicians specializing in prenatal diagnostics and fetal echocardiography. We divided CVAs into seven categories: septal defects, conotruncal anomalies, left-sided obstructive anomalies, right-sided obstructive anomalies, aortic arch anomalies, heterotaxy syndromes, and vascular anomalies (Table 1). Vascular anomalies included SUA, DV agenesis, PLSVC, and persistent right umbilical artery (PRUV). Cases of interrupted vena cava inferior (VCI) and anomalies in pulmonary vein drainage occurred in heterotaxy syndromes and were not reported separately.

We assumed that the detected anomalies were isolated anomalies if they were not accompanied by structural ECMs (the diagnosis was confirmed postnatally or in the case of pregnancy termination by means of autopsy) or intrauterine fetal growth restriction (FGR) prenatally. Minor extracardiac abnormalities, also referred to as ‘soft markers’, such as an echogenic bowel and choroid plexus cysts, were not treated as ECMs.

In 851 (84.7%) cases, invasive prenatal diagnostics were indicated because of abnormal results during Down syndrome screening (643 cases after the first-trimester scan and 208 cases after the second-trimester scan). In 38 (3.8%) cases, invasive tests were indicated because of advanced maternal age. In 116 (11.5%) cases, other reasons were given, such as parental request, family history of congenital defects, and genetic mutation carriership. Detailed indications for invasive testing and a flow chart showing genetic prenatal diagnosis in our department are shown in Figure 1.

Genetic testing (aCGH and G-band karyotyping) was performed at the Department of Medical Genetics, Jagiellonian University Medical College, following amniocentesis or chorionic villus sampling. Both aCGH and karyotyping were performed in 346 cases (in 330 cases due to abnormal ultrasound findings and in 16 cases due to parents having balanced chromosomal rearrangements), and aCGH alone was performed in 659 cases. Pre-test and post-test counseling was carried out by trained clinical geneticists. Those who chose to participate provided written informed consent after the discussion of the potential advantages and risks of chromosomal microarray testing, including the possibility of identifying variants of uncertain clinical significance (VUS) and genetic variants in the fetus that cause adult-onset disorders. Results of a pathological nature or the presence of VUS were revealed to patients, followed by comprehensive genetic counseling. In addition, in cases with fetal pathogenic and VUS CNVs, parents were encouraged to undergo aCGH to verify if the fetal CNVs were inherited or appeared de novo. However, to avoid potential iatrogenic effects in the parents, we did not recommend parental testing if a benign or likely benign CNV was found that was previously reported in the literature.

### 2.1. Array Comparative Genomic Hybridization Analysis and Interpretation

Unprocessed amniotic fluid (~10 mL) or chorionic villi were used for genomic DNA extraction using the Genomic Mini AX Body Fluids kit (A&A Biotechnology, Gdańsk, Poland) or QuickGene DNA tissue kit S (Kurabo Industries, Osaka, Japan). DNA concentration and purity were determined using a NanoDrop 1000 spectrophotometer (Thermo Scientific, Waltham, MA, USA). aCGH analysis was performed with the use of 8 × 60 K oligonucleotide microarrays (Agilent, Santa Clara, CA, USA). The two-color analysis required 200 ng of purified DNA. Test DNA and male reference DNA samples were labeled with Cy3 and Cy5 without previous digestion. The labeling and hybridization of the test and reference DNA were performed according to the oligonucleotide array-based CGH for genomic DNA analysis protocol. The scanning of the microarrays was performed using the Agilent SureScan Microarray Scanner. Data extraction, analysis, and visualization were performed using the CytoGenomics v 5.0 software. Each fetal DNA sample was treated as a sample of unknown sex and co-hybridized with male control DNA. If gender mismatch of the fetus was found in bioinformatics analysis in relation to the male reference DNA, this sample was re-analyzed in silico with the female reference DNA sample. The procedure of microarray analysis requires the establishment of a control field on the microarray slide, including female and male reference DNA. This approach made it possible to precisely detect the microdeletions and microduplications in the sex chromosomes even if the sex of the fetus was uncertain. A newly identified CNV was considered a relevant chromosomal aberration if it was detected in a minimum of ten consecutive oligonucleotide microarray probes. Aberrations were filtered up to a minimal size of 1 Mb for deletions and duplications. However, CNVs smaller than 1 Mb were included in further analysis if they were clearly pathogenic based on data from ClinVar (Clinically Relevant Variation Database; accessed 15 June 2022) and DECIPHER (Database of Chromosomal Imbalance and Phenotype in Humans Using Ensembl Resource, http://decipher.sanger.ac.uk/; accessed 2 May 2022), databases supported by data from OMIM (Online Mendelian Inheritance in Man: http://www.ncbi.nlm.nih; accessed 23 June 2022), and the UniQue databases (accessed 2 May 2022). The CNVs were assigned the following interpretations according to international guidelines [31,32,33,34]: (1) pathogenic or likely pathogenic, (2) variants of uncertain significance (VUS), or (3) benign or likely benign.

The institutional review board waived the requirement for a separate ethical approval for this analysis, since the interview, invasive procedures, genetic testing, and sonographic evaluations were performed as integral parts of routine clinical care, for which informed consent had been previously given by the women.

Additionally, as part of the research, dedicated software was developed to collect and store the data, with the requirement of data anonymization, as well as to prepare the data for statistical analysis.

### 2.2. Statistical Analysis

Patient characteristics are described as means with standard deviation for normally distributed numerical data and as percentages for categorical variables. The Shapiro–Wilk test was used to test the normality of distribution. Differences were analyzed by Student’s *t*-test for normally distributed data and by the Mann–Whitney U-test for non-normally distributed data. Chi-square and Fisher’s exact tests were used for comparisons of categorical variables. In all analyses, *p* values < 0.05 were considered statistically significant.

## 3. Results

### 3.1. Ultrasound Findings

Among 1005 women referred to invasive prenatal testing, 202 (20.0%) cases of CVAs were detected. CVAs were the most common anomalies that occurred in the studied population. Congenital heart defects were present in 168 cases, and in 34 cases CHDs were associated with vascular anomalies (Figure 2). In another 34 cases, heart anatomy was normal but vascular anomalies were detected. In one case, both SUA and LSVC were identified. A detailed list of cardiac and vascular anomalies is presented in Table 1. The most common cardiac anomalies were septal defects followed by conotruncal anomalies, which affected 94 (46.5%) and 38 (18.8%) of the fetuses with CVAs, respectively (Table 1). The most common vascular anomaly was SUA, which was detected in 24.7% of fetuses with CVAs (Table 1). The groups with and without CVAs differed significantly regarding first-trimester markers such as NT, TR, DV blood flow, and NB hypoplasia, as they were more frequently abnormal in the CVA group (Table 2). In 32.9% of cases with CVAs, NT was above 3.0 mm.

In the CVA group, accompanying ECMs occurred more frequently than in fetuses without CVAs (Table 2). In 60.9% of cases with CVAs, one or more associated ECMs were detected (Table 2). The most common associated anomalies were anomalies of the face and neck followed by skeletal defects, whereas in the group without CVAs, the most frequent ECMs were skeletal and central nervous system anomalies (Table 2).

### 3.2. Cytogenetic Analysis

In the group of fetuses with CVAs, pathogenic chromosomal aberrations were found in 47.5% (96/202) of cases. The aberrations were more substantially frequent than in the group of fetuses without CVAs (80/803; 9.9%), and the difference was statistically very significant (*p ≤* 0.00004) (Table 2). In fetuses with CVAs, numerical chromosomal aberrations accounted for 38.6% (78/202) of cases, of which the most common was trisomy 21 (48.7%), followed by trisomy 18 (25.6%). In one case, trisomy 18 coexisted with trisomy 13, and in another case, de novo translocation was detected (46,XY,t(6;14)(p25.3;q11.2)). In the remaining 124 fetuses with CVAs, the number of chromosomes was normal, but in 27 cases (21.7%; 27/124), aCGH revealed CNVs, of which 18 (14.5%; 18/124) were classified as pathogenic, 6 (4.8%; 6/124) as VUS CNVs, and 3 (2.4%; 3/124) as benign CNVs (Figure 1). The size of the CNVs varied between 0.1 Mb and 56.7 Mb. The 22q11.2 microdeletion was the most common CNV, as it occurred in 4.8% of euploid fetuses (6/124), encompassing 33.3% (6/18) of all pathogenic CNVs detected. A detailed list of CNVs and associated ultrasound findings is presented in Table 3. Pathogenic CNVs were more frequent in the group with CVAs (18 cases: 8.9%) than in the remaining fetuses (25 cases: 3.1%). The difference was statistically significant (*p* = 0.0003) (Table 2). The differences between the groups with and without CVAs regarding VUS and benign CNVs did not reach statistical significance.

Septal defects were the most common heart defects among fetuses with numerical chromosomal aberrations (67.9%; 53/78) (Table 4), while conotruncal defects were the most common in the group of fetuses with pathogenic CNVs (33.3%; 6/18) (Table 3).

Pathogenic CNVs were more frequently detected when associated with ECMs (Figure 2; Table 3). In fetuses with isolated CHDs, chromosomal aberrations occurred in 17.6% (9/51) of cases, and of this group pathogenic CNVs were detected in 3/51 (5.8%). When CHDs were associated with vascular and other ECMs, chromosomal aberrations were observed in 76.0% (89/117), including 13/117 (11.1%) in which the aberration was a pathogenic CNV (Figure 2).

In the group with pathogenic CNVs, eight (44.4%; 8/18) women opted for termination, three (16.6%; 3/18) fetuses died before birth, and seven were born, but five (27.7%; 5/18) died after delivery and two (11.1%; 2/18) survived.

In the group of fetuses with trisomy 21, parents opted for the termination of pregnancy in 25 (65.7%) cases, no intrauterine fetal demise (IUFD) was recorded, and 13 were live-born. Termination was also opted for in 17 (85.0%) cases with trisomy 18, in 7 (100.0%) cases with trisomy 13, and in 5 (71.4%) cases with triploidy. IUFD was diagnosed in three (15.0%) cases of trisomy 18, one (20.0%) case of Turner syndrome, and two (28.5%) cases of triploidy. In total, in 65.2% (62/95) of cases of CVAs with confirmed chromosomal aberrations, parents opted for termination. On the contrary, this only happened in 6.5% (7/107) of cases of fetuses with CVAs without genetic defects. Moreover, fetuses with a chromosomal anomaly died intrauterine approximately twice as often as those without such an anomaly (9/95 cases vs. 5/107 cases; 9.4% vs. 4.6%, respectively).

## 4. Discussion

In this retrospective cohort study of more than 1000 fetuses undergoing invasive prenatal testing, we looked in detail at cardiovascular anomalies, associated non-cardiac structural defects, and chromosomal aberrations.

Our results confirm previous observations [35] that CVAs are the most common anomalies in fetuses, as they were revealed in 20% of the studied cohort, and that they are often associated with ECMs [10,11]. In our study, as well as in a study by Zhu et al. [36], septal defects were the most common heart anomalies identified. However, septal defects predominated among fetuses with numerical aberrations, while conotruncal defects were the most common findings among fetuses with pathogenic CNVs. In general, unbalanced chromosomal aberrations accompanied CVAs in 47.5% of cases, which is more than previously reported. In the study by Kowalczyk et al. [37], the aCGH technique revealed aneuploidies and structural aberrations in 37% of fetuses with heart anomalies. In our study, the most common chromosomal aberrations associated with CVAs were trisomies 21, 18, and 13, and triploidy, while the Pediatric Cardiac Genomics Consortium did not report trisomies 18 and 13 and triploidy as common findings [7]. This difference could be due to the fact that most pregnancies that are complicated by trisomies 18 and 13 and triploidy are terminated, or fetuses die in utero and are not reported postnatally. It should also be stated that this is a population selected based on a prenatal suspicion of abnormalities. Thus, differences in comparison with other studies may result from methodological differences, in particular, differences in ultrasound expertise and the experience of the doctors conducting the examination.

It was shown that the CMA technique in prenatal diagnostics demonstrates an additional diagnostic yield of pathogenic findings between 1.7% and 9% if the karyotype is normal [21,22,23] and about 10% in fetuses with multiple anomalies detected by ultrasound [38]. However, it was underlined that, depending on the selection criteria, these numbers may be higher, especially in fetuses with multiple anomalies such as cardiac and renal anomalies [39,40].

We detected pathogenic CNVs in 14.5% of cases with CVAs and normal karyotype, which is more than previously reported. Yan et al. [41] reported pathogenic CNVs among fetuses with a normal karyotype and without 22q11.2 microdeletion in 6.6% of cases, whereas in our study, it was 9.6%. Meta-analysis by Jansen et al. [15] showed an additional 7% of CNVs after excluding aneuploidies and 22q11.2 microdeletion. Additionally, taking into account VUS, our results indicate a higher incidence than in the study by Jansen et al. [15] (4.8% vs. 3.4%). In turn, in postnatal cohorts of isolated CHD without aneuploidies and 22q11 microdeletion, the percentages of CNVs were reported to be 0–4% [41,42,43]. This discrepancy can be attributed to non-comparable cohorts and the fact that prenatal ultrasound does not detect all dysmorphic features or other subtle symptoms of genetic syndromes, which can be detected postnatally. Although most cardiac defects can be diagnosed prenatally by fetal echocardiography, many anomalies are not identified in routine prenatal ultrasound. American studies reported only ≈30% of fetuses with serious CHDs being identified prenatally, whereas, in Sweden, this proportion reached 56% [44,45].

On the other hand, some fetuses die before birth, and in some cases, the parents choose to terminate the pregnancy. Both in our study and in the observations of other authors [46,47], parents more often chose the option of termination in the case of diagnosed chromosomal aberrations. All this may lead to an underestimation of the diagnostic yield of chromosomal aberrations in cohorts of fetuses with cardiovascular defects. In our study, the cohort consisted of fetuses in the first trimester and at the beginning of the second trimester, so before parents had considered terminating the pregnancy.

Interestingly, in our observation, heterotaxy syndromes, which are not considered to be associated with large chromosomal anomalies detected by karyotyping, were found to have pathogenic CNVs in two cases and, in one case, left isomerism coexisted with trisomy 21. Thus, the availability of more advanced techniques, such as aCGH, allows the detection of the genetic cause of defects that, until now, were considered not to coexist with chromosomal aberrations.

It is also worth noting that the most common vascular anomaly, SUA, was associated with chromosomal aberrations in 52% of cases, which was also described by other authors [48].

It should be emphasized that due to the limitations of aCGH, further diagnostic tests can be considered if a cardiovascular defect is prenatally detected, such as next-generation sequencing-based approaches: clinical exome sequencing (CES), whole exome sequencing (WES), or whole genome sequencing (WGS) [36,49]. However, the above tests are available in our center only on a commercial basis and are not reimbursed by the national health care system. According to recent findings, the prenatal use of exome sequencing analysis in cases with multiple ultrasound-detected anomalies, especially skeletal anomalies and cardiac defects, showed a higher diagnostic yield (15.4%) over conventional testing [50]. Nevertheless, it is important to underline that even when karyotyping, CMA analysis and next-generation sequencing approaches are applied, we may not find a causal genetic variant in all cases [49].

It should also be emphasized that in the group of fetuses with an abnormal aneuploidy screening result, and in the case of other structural defects, a detailed assessment of the fetal heart should be recommended prenatally. We believe that the detection of a cardiovascular defect should be followed by comprehensive genetic testing, optimally based on the aCGH technique or a combination of aCGH and karyotyping. This procedure allowed seven cases of triploidy and one case of a translocation to be diagnosed.

## 5. Limitations

There are some limitations to our study. First, the sample size was relatively small and came from a single referral center. Moreover, the follow-up mainly covered the pregnancy until termination or delivery, and postnatal observation was limited. Furthermore, it should also be emphasized that a range of additional, clinically relevant genetic diagnoses could possibly be detected if using other molecular techniques such as next-generation sequencing.

## 6. Conclusions

In conclusion, this cohort study shows that CVAs are the most common malformations detected in fetuses undergoing invasive prenatal testing. Approximately every second affected fetus may have chromosomal aberration. After an aneuploidy is excluded, pathogenic copy number variants are identified with the use of aCGH in a substantial proportion of cases with cardiovascular anomalies. aCGH should, therefore, be considered as a diagnostic method of choice for fetuses with CVAs detected by ultrasound examination, especially if accompanied by other structural malformations.

## Figures and Tables

**Figure 1 ijerph-19-10019-f001:**
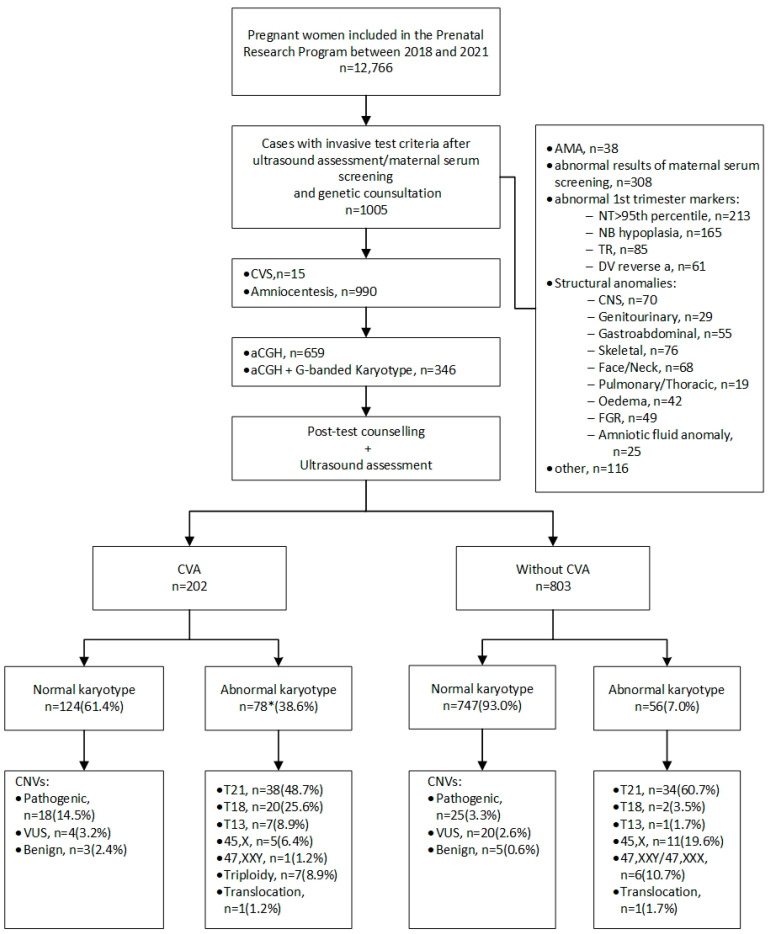
Flowchart showing genetic prenatal diagnosis in the Department of Obstetrics and Perinatology, Kraków, Poland. aCGH—array comparative genomic hybridization; AMA—advanced maternal age; CNS—central nervous system; CNVs—copy number variants; CVA—cardiovascular anomaly; CVS—chorionic villus sampling; DV—ductus venosus; NB—nasal bone; NT—nuchal translucency; TR—tricuspid regurgitation; T13—trisomy 13; T18—trisomy 18; T21—trisomy 21; VUS—variants of uncertain clinical significance. * In one case, trisomy 18 coexisted with trisomy 13.

**Figure 2 ijerph-19-10019-f002:**
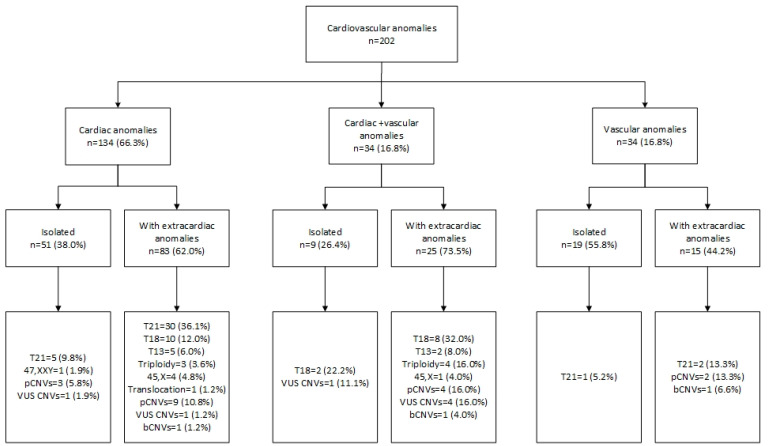
Cardiovascular anomalies and associated genetic findings. bCNVs—benign copy number variants; pCNVs—pathogenic copy number variants; T13—trisomy 13; T18—trisomy 18; T21—trisomy 21; VUS—variants of uncertain clinical significance; 45,X—Turner syndrome.

**Table 1 ijerph-19-10019-t001:** Diagnostic types of cardiovascular anomalies detected among 1005 pregnant women referred for invasive prenatal testing.

	Total, *N* = 202	Associated ECMs, *N*(%)	Chromosomal Aberrations, *N*(%)
Septal Defects, *n*(%)	94(46.5)	62(65.9)	66(70.2)
– AVSD simple, *n*(%)	22(10.8)	18(81.8)	21(95.4)
– Simple VSD, *n*(%)	59(29.2)	39(66.1)	36(61.0)
– Septal defect + other CVAs, *n*(%)	13(6.4)	5(38.4)	9(69.2)
Conotruncal anomalies, *n*(%)	38(18.8)	18(47.3)	19(50.0)
– DORV, *n*(%)	19(9.4)	11(57.8)	11(57.8)
– d-TGA simple, *n*(%)	3(1.4)	1(33.3)	0(0.0)
– TOF simple, *n*(%)	10(4.9)	3(30.0)	5(50.0)
– Complex conotruncal anomalies, *n*(%)	6(2.9)	3(50.0)	3(50.0)
Left-sided obstructive anomalies, *n*(%)	25(12.3)	13(52.0)	13(52.0)
– HLHS, *n*(%)	5(2.4)	2(40.0)	2(40.0)
– CoA, *n*(%)	13(6.4)	6(46.1)	6(46.1)
– CoA + septal defect, *n*(%)	5(2.4)	4(80.0)	4(80.0)
– IAA + VSD, *n*(%)	1(0.4)	1(100.0)	1(100.0)
– AS, *n*(%)	1(0.4)	0(0.0)	0(0.0)
Aortic arch anomalies, *n*(%)	20(9.9)	13(65.0)	6(30.0)
– RAA total, *n*(%)	12(5.9)	7(63.6)	2(18.1)
– DAA, *n*(%)	1(0.4)	0(0.0)	1(100.0)
– ARSA total, *n*(%)	7(3.4)	6(85.7)	3(42.8)
Right-sided obstructive anomalies, *n*(%)	8(3.9)	2(25.0)	5(62.5)
– HRHS, *n*(%)	4(1.9)	1(25.0)	3(75.0)
– Ebstein anomaly, *n*(%)	1(0.4)	1(100.0)	1(100.0)
– PV dysplasia, *n*(%)	3(1.4)	0(0.0)	1(33.3)
Fetal heterotaxy, *n*(%):	8(3.9)	8(100.0)	3(37.5)
– Left atrial isomerism, *n*(%)	6(2.9)	6(100.0)	2(33.3)
– Right atrial isomerism, *n*(%)	2(0.9)	2(100.0)	1(50.0)
SUA total, *n*(%)	50(24.7)	31(62.0)	26(52.0)
PLSVC total, *n*(%)	10(4.9)	6(60.0)	4(40.0)
DV agenesis, *n*(%)	8(3.9)	0(0.0)	0(0.0)
PRUV, *n*(%)	1(0.4)	0(0.0)	0(0.0)

ARSA—aberrant right subclavian artery; AS—aortic stenosis; AVSD—atrioventricular septal defect; CoA—coarctation of the aorta; CVAs—cardiovascular anomalies; DAA—double aortic arch; DORV—double-outlet right ventricle; DV—ductus venosus; ECMs—extracardiac malformations; HLHS—hypoplastic left heart syndrome; HRHS—hypoplastic right heart syndrome; IAA—interrupted aortic arch; PLSVC—persistent left superior vena cava; d-TGA—transposition of the great arteries; PRUV—persistent right umbilical artery; PV—pulmonary valve; RAA—right aortic arch; SUA—single umbilical artery; TOF—tetralogy of Fallot; VSD—ventricular septal defect.

**Table 2 ijerph-19-10019-t002:** Numerical chromosomal aberrations, copy number variants, and structural anomalies among 1005 fetuses with and without cardiovascular anomalies (CVAs).

	CVAs *N* = 202 (20.1)	Without CVAs *N* = 803 (79.9)	P
Maternal age, years			
Mean ± SD	31.9 ± 5.4	32.9 ± 5.3	0.0309
Min–max	17–45	18–48	
Median	32	33	
Male, *n*(%)	120(59.4)	433(53.9)	0.1614
Pathogenic chromosomal aberrations, *n*(%)	96(47.5) *	80(9.9)	0.0000
Trisomy 21, *n*(%)	38(18.8)	34(4.2)	0.0000
Trisomy 18, *n*(%)	20(9.9)	2(0.2)	0.0000
Trisomy 13, *n*(%)	7(3.4)	1(0.1)	0.0000
Turner syndrome, *n*(%)	5(2.4)	11(1.3)	0.2619
47,XXX; 47,XXY, *n*(%)	1(0.5)	6(0.7)	0.7001
Triploidy, *n*(%)	7(3.4)	0(0.0)	-
Translocation, *n*(%)	1(0.5)	1(0.1)	-
CNVs:			
– Pathogenic CNVs, *n*(%)	18(8.9)	25(3.1)	0.0003
– VUS CNVs, *n*(%)	6(2.9)	20(2.5)	0.4438
– Benign CNVs, *n*(%)	3(1.5)	5(0.6)	0.2176
Associated structural anomalies, *n*(%)			
– None, *n*(%)	79(39.1)	596(74.2)	0.0000
– One or more anomalies, *n*(%)	123(60.9)	208(25.8)	
Structural anomalies,			
– CNS, *n*(%)	26(12.8)	44(5.5)	0.0002
– Genitourinary tract, *n*(%)	16(7.9)	13(1.6)	0.0000
– Gastroabdominal, *n*(%)	30(14.8)	25(3.1)	0.0000
– Skeletal, *n*(%)	31(15.3)	45(5.6)	0.0000
– Face/neck, *n*(%)	38(18.8)	30(3.7)	0.0000
– Pulmonary/thoracic, *n*(%)	12(5.9)	7(0.8)	0.0000
– Oedema, *n*(%)	16(7.9)	26(3.2)	0.0029
– FGR, *n*(%)	26(12.8)	23(2.9)	0.0000
– Amniotic fluid anomaly, *n*(%)	13(6.4)	12(1.5)	0.0002
First trimester markers:			
– NT ≥ 3.5 mm, *n*(%)	44(25.9)	78(10.3)	0.0000
– NT > 3.0 mm, *n*(%)	56(32.9)	113(14.9)	0.0000
– TR, *n*(%)	34(16.8)	51(6.3)	0.0000
– DV abnormal, *n*(%)	24(11.8)	37(4.6)	0.0001
– NB hypoplasia, *n*(%)	50(24.8)	115(14.3)	0.0004

CNS—central nervous system; CNVs—copy number variants; CVAs—cardiovascular anomalies; DV—ductus venosus; FGR—fetal growth restriction; NB—nasal bone; NT—nuchal translucency; SD—standard deviation; TR—tricuspid regurgitation; VUS— variants of uncertain clinical significance. * Trisomy 18 and 13 coexisted in one case.

**Table 3 ijerph-19-10019-t003:** Copy number variants found by aCGH in fetuses with cardiovascular anomalies among 1005 pregnant women undergoing invasive prenatal testing.

Case	Microarray Nomenclature	Size (Mb)	Cardiac/Vascular Defect	Extra-Cardiac Defect	Pathogenicity Classification	Obstetrical Outcomes
1	arr[GRCh37]1p36.32p35.3(2558854_29403494)x3	26.8	DORV TOF type	Oedema	P	IUFD
2	arr[GRCh37]2q13(110862477_110964737)x1	0.1	Multiple VSDs	Nasal hypoplasia	B	TOP
3	arr[GRCh37]2p16.3(51137071_51382872)x1	0.24	DILV TGA VSD PS	-	P	TOP
4	arr[GRCh37]3q29(194969955_197317103)x1	2.35	PV dysplasia	-	P	TOP
5	arr[GRCh37]3q23q29(141143997_197837049)x3	56.7	DORV	Club feet	P	Born Died 2 months after surgery
6	arr[GRCh37]4q35.1q35.2(186737240_188908875)x1	2.1	DAA	Megacysterna magna	VUS	Alive and well
7	arr[GRCh37]4q32.3(165560826_166112340)x3	0.55	VSD	-	VUS	Alive and well
8	arr[GRCh37]5p12(44506416_45414642)x3	0.9	DORV SUA	Omphalocele	VUS	TOP
9	arr[GRCh37]5p15.33p15.2(22149_10044258)x1	10	DORV TOF type	-	P	Born Died
10	arr[GRCh37]5p15.33p15.31(22149_7449397)x1,5p15.31p12(7506131_44341490)x3	7.4 36.84	VSD PLSVC	Nasal hypoplasia Left-sided hydronephrosis	P P	Born Died after 7th day of life
11	arr[GRCh37]7q31.2q32.2(116255056_129694097)x3,7q32.2q36.3(129853288_159085681)x1	13.429.2	VSD	Cleft palate FGR CCA	P VUS	IUFD
12	arr[GRCh37] 8p12p11.23(35930461_37002743)x3 mos	1.1	CoA SUA	Hypospadias FGR Nasal hypoplasia	VUS	Alive and well
13	arr[GRCh37]9p24.3p13.3(204193_34973544)x3, 18p11.32p11.21(148963_11218383)x1	35.811.1	SUA	Cleft palate hydrocephalus	P	TOP
14	arr[GRCh37]17q25.1q25.3(68451323_78644236)x3	10.2	SUA	Nasal hypoplasia cerebellar vermis hypoplasia	P	TOP
15	arr[GRCh37]20p11.23p11.21(19852522_22840889)x1	2.99	Left isomerism (AVSD interrupted VCI with azygous continuation bradycardia)	Situs inversus	P	TOP
16	arr[GRCh37]21q21.1(23663169_23927226)x1	0.26	TOF SUA	CDH cerebellar vermis hypoplasia Nasal hypoplasia CCA Hypospadias	B	TOP
17	arr[GRCh37]22q12.3(34043679_34212026)x1	0.17	HLHS	Cerebellar hypoplasia Hands and forearms hypoplasia Facial dysmorphia	VUS	Born Died 4 days after birth
18	arr[GRCh37]22q11.21(18894835_21505417)x1	2.6	TOF PLSVC	Thymus agenesis	P	TOP
19	arr[GRCh37]22q11.1q11.21(17397498_20311763)x1	2.9	Right isomerism (AVSD, DORV, TAPVC) SUA	Thymus aplasia	P	Born Died after 2 months
20	arr[GRCh37]22q11.21(18661724_21440514)x1	2.77	HRHS	Nasal hypoplasia	P	TOP
21	arr[GRCh37]22q11.21(18904835_21505400)x1	2.6	IAA VSD	Thymic hypoplasia	P	TOP
22	arr[GRCh37]22q11.21q11.22(21808950_22905068)x1	1.1	Multiple VSDs	Cleft lip and palate	P	PPROM, premature birth at 29 weeks, Alive
23	arr[GRCh]22q11.21(18761724_21470514)x1	2.7	AVSD TOF LSVC		P	Born Died 2 months after surgery
24	arr[GRCh37]Xp21.3(26241961_26607442)x0, 11p14.2p14.1(27006061_27225374)x3	0.36 0.22	TA, PS VSD Coronary artery fistula SUA		VUS B	PPROM at 30 weeks of pregnancy Died
25	arr[GRCh37]Xp22.31(6488721_8097511)x2	1.6	SUA	Gastroschisis	B	Born Died 2 months after surgery
26	arr[GRCh37]Xp22.33(169796_2778489)x0, Xq21.31q21.32(88453553_92297659)x0,	2.61 3.84	CoA VSD	Hhydropericardium nuchal oedema	P	IUFD
27	arr[GRCh37]Xq28(149116213_155232907)x1	6.1	VSD RAA	Thymic hypoplasia	P	Alive and well

aCGH—array comparative genomic hybridization; AVSD—atrioventricular canal; B—benign; CCA—corpus callosum agenesis; CDH—congenital diaphragmatic hernia; CoA—coarctation of the aorta; DAA—double aortic arch; DILV—double-inlet left ventricle; DORV—double-outlet right ventricle; FGR—fetal growth restriction; HLHS—hypoplastic left heart syndrome; HRHS—hypoplastic right heart syndrome; IAA—interrupted aortic arch; IUFD—intrauterine fetal demise; P—pathogenic; PLSVC—persistent left superior vena cava; PS—pulmonary stenosis; RAA—right aortic arch; SUA—single umbilical artery; TA—tricuspid valve atresia; TGA—transposition of the great arteries; TOF—tetralogy of Fallot; TOP—termination of pregnancy; VUS—variants of uncertain clinical significance; VSD—ventricular septal defect.

**Table 4 ijerph-19-10019-t004:** Cardiovascular anomalies and numerical chromosomal aberrations in 1005 pregnant women undergoing invasive prenatal testing.

	Trisomy 21, *N* = 38	Trisomy 18, *N* = 20	Trisomy 13, *N* = 7	Triploidy, *N* = 7	Turner Syndrome, *N* = 5	47, XXY *N* = 1	De Novo Translocation *N* = 1	Total, *N* = 78 *
VSD, *n*(%)	13(34.2)	11(55.0)	3(42.8)	3(42.8)	0	1(100.0)	1(100)	32(41.0)
AVSD, *n*(%)	15(39.5)	5(25.0)	1(14.2)	0	0	0	0	21(26.9)
TOF, *n*(%)	2(5.2)	0	0	0	0	0	0	2(2.5)
DORV, *n*(%)	1(2.6)	3(15.0)	2(28.5)	2(28.5)	0	0	0	8(10.2)
CoA, *n*(%)	0(0.0)	0	1(14.2)	0	4(80.0)	0	0	5(6.4)
CoA + VSD, *n*(%)	1(2.6)	1(5.0)	0	0	1(20.0)	0	0	3(3.8)
HLHS, *n*(%)	1(2.6)	0	0	0	0	0	0	1(1.2)
Ebstein anomaly, *n*(%)	1(2.6)	0	0	0	0	0	0	1(1.2)
– Left isomerism, *n*(%)	1(2.6)	0	0	0	0	0	0	1(1.2)
PA, *n*(%)	1(2.6)	0	0	0	0	0	0	1(1.2)
Other:								
– ARSA, *n*(%)	1(2.6)	0	1(14.2)	0	0	0	0	2(2.5)
– RAA, *n*(%)	0	0	0	1(14.2)	0	0	0	1(1.2)
– SUA, *n*(%)	3(7.8)	3(15.0)	1(14.2)	2(28.5)	1(20.0)	0	0	10(12.8)
– PLSVC, *n*(%)	0	1(5.0)	0	1(14.2)	0	0	0	2(2.5)

* Trisomy 18 and 13 were comorbid in one case. ARSA—aberrant right subclavian artery; AVSD—atrioventricular septal defect; CoA—coarctation of the aorta; DORV—double-outlet right ventricle; HLHS—hypoplastic left heart syndrome; PA—pulmonary atresia; PLSVC—persistent left superior vena cava; RAA—right aortic arch; SUA—single umbilical artery; VSD—ventricular septal defect.

## Data Availability

The data presented in this study are available on request from the corresponding author. The data are not publicly available due to privacy/ethical reason.

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
