# Peer review of "Cardiovascular Anomalies among 1005 Fetuses Referred to Invasive Prenatal Testing—A Comprehensive Cohort Study of Associated Chromosomal Aberrations"

_ijerph, 2022, doi:10.3390/ijerph191610019_

Round 1
Reviewer 1 Report
I was involved in the assessment of the former version of the paper, and have checked the comments and subsequent adaptations made.
I still recommend to reconsider the abstract. The second sentence (on 20 % of cases) cannot be understood if you do no provide additional information on the activity (cases seen ?) of your unit. Either provide additional information, or remove this sentence ? Or do the 1005 cases reflect all cases, of whom 20 % this diagnostic intervention was based on a CVA observed ? Were the ECM based ‘only’ on prenatal, or also on postnatal assessment ? Please also reconsider the last sentence, as the cardiac malformation was the initiator, why do you subsequent only stress the need to assess the cardiovascular anatomy, and not the ECM ?
The flow chart is indeed of add on value.
A6 comment: not sure if I understand this correct, were benign and likely benign CNV not considered for parental testing (if so, how to discriminate between de novo fetal and already present CNV), or how has this been addressed ? or were these findings not discussed with the parents.
The paper does not yet respect the guidelines of the journal (author contributions, funding, institutional review board statement, informed consent statement, data availability statement, acknowledgements, conflicts of interest
Minor, textual
Reimbursable, should likely read reimbursed ?
Author Response
Response to the Reviewer 1
We are grateful for your insightful comments on our paper. We incorporated some changes to reflect the suggestions provided. We have highlighted the changes within the manuscript.
Comments and Suggestions for Authors:
I was involved in the assessment of the former version of the paper, and have checked the comments and subsequent adaptations made.
I still recommend to reconsider the abstract. The second sentence (on 20 % of cases) cannot be understood if you do no provide additional information on the activity (cases seen ?) of your unit. Either provide additional information, or remove this sentence ? Or do the 1005 cases reflect all cases, of whom 20 % this diagnostic intervention was based on a CVA observed ? Were the ECM based ‘only’ on prenatal, or also on postnatal assessment ? Please also reconsider the last sentence, as the cardiac malformation was the initiator, why do you subsequent only stress the need to assess the cardiovascular anatomy, and not the ECM ?
Re: We included the suggested changes in Abstract.
The flow chart is indeed of add on value.
A6 comment: not sure if I understand this correct, were benign and likely benign CNV not considered for parental testing (if so, how to discriminate between de novo fetal and already present CNV), or how has this been addressed ? or were these findings not discussed with the parents.
Re: We modified the sentence as follows:
However, to avoid potential iatrogenic effects in the parents, we did not recommend parental testing if a benign or likely benign CNV was found that was previously reported in the literature.
The paper does not yet respect the guidelines of the journal (author contributions, funding, institutional review board statement, informed consent statement, data availability statement, acknowledgements, conflicts of interest –
Re: We addressed the above issues in the submission system.
Minor, textual
Reimbursable, should likely read reimbursed ?
Re: We changed to reimbursed.
Reviewer 2 Report
This is a useful study, going into considerable detail about the actual type of CVA that can be associated with a chromosome abnormality, classic or CNV. It was of interest that the type of CVA, septal or conotruncal, showed a correlation with the prenatal test type, classic or CNV.
As the aa. point out, the numbers are small, in some categories (Table 4) n = 1. This is not a criticism, merely an observation. These are still good data.
Of course it’s obvious, but I think, nevertheless, it would be worth stating explicitly that this is a selected population: selected upon the basis of a prenatal suspicion of an abnormality. So differences (such as they are) in comparison with other studies may reflect differences in methodology, and in particular, variation in ultrasonographic expertise.
Could the aa. clarify their somewhat "stand-alone" comments about further methodologies: CES, WES, WGS. They say this should be considered; but did not do so in this present study. Why not? Do they plan to add these tests in the future?
I have made a few very minor comments/suggestions on the draft, which the aa. might wish to consider.

Author Response
Response to the Reviewer 2
We are grateful for your insightful comments on our paper. We incorporated some changes to reflect the suggestions provided. We have highlighted the changes within the manuscript.
Comments and Suggestions for Authors
This is a useful study, going into considerable detail about the actual type of CVA that can be associated with a chromosome abnormality, classic or CNV. It was of interest that the type of CVA, septal or conotruncal, showed a correlation with the prenatal test type, classic or CNV.
As the aa. point out, the numbers are small, in some categories (Table 4) n = 1. This is not a criticism, merely an observation. These are still good data.
Of course it’s obvious, but I think, nevertheless, it would be worth stating explicitly that this is a selected population: selected upon the basis of a prenatal suspicion of an abnormality. So differences (such as they are) in comparison with other studies may reflect differences in methodology, and in particular, variation in ultrasonographic expertise.
Re: We added in the Discussion following statement:
It should also be stated that this is a population selected based on prenatal suspicion of abnormalities. Thus, differences in comparison with other studies may result from methodological differences, in particular differences in ultrasound expertise and experience of the doctors conducting the examination.
Could the aa. clarify their somewhat "stand-alone" comments about further methodologies: CES, WES, WGS. They say this should be considered; but did not do so in this present study. Why not? Do they plan to add these tests in the future?
RE: Currently, in our department, the parents are informed about the possibility of performing WES or WGS, which are available in Poland on a commercial basis and are not reimbursed by the national health care system. We comment on this issue in the text of the manuscript.
I have made a few very minor comments/suggestions on the draft, which the aa. might wish to consider.
Re: Thank you for your comments, we have included them in our paper.
This manuscript is a resubmission of an earlier submission. The following is a list of the peer review reports and author responses from that submission.
Round 1
Reviewer 1 Report
This manuscript has analyzed the ultrasonic cardiovascular anomalies associated with the numerical chromosomal abnormalities and copy number variants (CNVs) among 1005 fetuses referred to invasive prenatal testing. The results can provide a valuable view for clinical practice while some of papers and reviews already reported them. Now there are some concerns to be addressed.
1. In the session of material and methods, please list the detailed indications for these 1005 fetuses referred to invasive prenatal testing. Based on the presentation of line 97-98, did the indications include the other ultrasonic anomalies besides abnormal Nuchal translucency (NT)? Had every case been performed ultrasound examination and G-band karyotyping? or G-band karyotyping only in the case with ultrasonic anomalies? And performed in which phase of the study? Please generate a flow diagram of this study and give the relevant details/classification in every phase so as to organize the materials and methods of this study clearly.
2. In this study, the aCGH was performed by Aglient 8x60k microarray and only male reference DNA. Why not female reference DNA included? If gender mismatch for female sample in only male reference DNA in Agilent platforms, the microdeletion or microduplication in the sex chromosome could be missed in the software or even manual analyses data.
3. The parental study was mentioned in the line 105-106. Did it also go for the fetus with benign CNV?
4. In the part of results, the data analyses with many kinds of percentages in different groups were applied for results presentation, but the detailed calculations for these percentages in relevant groups were shown incompletely in the text or table or figure, even no clear statement for the same data with two different percentages, such as: two different percentages for the same CNVs data in the table 2 and in line 214-215, but no calculations for them in the table/text, and so on. The reader is easy to be confused for these results.
5. Some other issues are in the part of results.
a) The title of figure 1 was “Cardiovascular anomalies and associated genetic disorders”. However, benign CNV is not genetic disorders.
b) In table 1, why the data for “Aortic arch anomalies” part and last part were shown differently from other parts?
c) In table 2, the case number for “Any autosomal or sex-chromosome Abnormality” was 77. However, if adding the case numbers from “Trisomy 21” to “Triploidy”, the number was 78. Furth more, there was one case with translocation, which was mentioned in line 322-323 in the discussion. But this translocation case was not shown in table 2 or the text of result.
6. Prevalence is the rate of all cases (new and pre-existing cases) number of a disease in a specific population at a particular timepoint or over a specified period of time. Incidence is the rate of new cases of a disease occurring in a specific population over a particular period of time. In this study, it should be better to use the terms of diagnostic yield/detection rate rather than prevalence/incidence. Also, some of medical genetic terms were inconsistent or incorrect in this manuscript. For example, “genetic disorders”, “genetic anomalies”, “genetic defects” and “genetic abnormalities” all came out in this manuscript as well as “numerical chromosomal aberrations/ numerical chromosomal anomalies/ numerical chromosomal abnormalities”. In line 95, “maternal age” should be “advanced maternal age”. So, please refer to "Thompson & Thompson Genetics in Medicine" for the terminology.
7. Please edit English language and style in this manuscript.
Author Response
Response to the Reviewer 1
We are grateful for your insightful comments on our paper. We have been able to incorporate changes to reflect most of the suggestions provided. We have highlighted the changes within the manuscript. Here is a point-by-point response to the reviewer' comments and concerns.
- In the session of material and methods, please list the detailed indications for these 1005 fetuses referred to invasive prenatal testing. Based on the presentation of line 97-98, did the indications include the other ultrasonic anomalies besides abnormal Nuchal translucency (NT)? Had every case been performed ultrasound examination and G-band karyotyping? or G-band karyotyping only in the case with ultrasonic anomalies? And performed in which phase of the study? Please generate a flow diagram of this study and give the relevant details/classification in every phase so as to organize the materials and methods of this study clearly.
RE: We have generated a flow diagram of this study and added as a Figure 1.
Also in the Materials and Methods section we clarified:
Ultrasound examination was performed in each case before and after invasive testing. First-trimester screening and second-trimester screening were performed following Polish and international guidelines [33,34]. Nuchal translucency (NT) above the 95th percentile was considered abnormal. Additionally, the ultrasound assessment included blood flow through the tricuspid valve (TV), DV, presence/absence of nasal bone (NB), and anatomy of the fetuses. During first-trimester screening, in each case, serum markers such as free β-human chorionic gonadotrophin (free β-hCG) and pregnancy associated plasma protein A (PAPP-A) were measured according to the guidelines [33,34]. Each patient was also informed about the possibility of non-invasive prenatal genetic testing (NIPT) as a screening method but that these tests are not reimbursable by the Polish National Health Fund and were performed in only 21 cases. Second trimester screening was based on ultrasound only.
Since 2018, in our Department, in cooperation with Department of Medical Genetics, Jagiellonian University Medical College, aCGH has been offered as a first-line test for prenatal testing and in cases with ultrasound detected anomalies. aCGH with G-band karyotyping was offered in 330 cases (n=330). Additionally, in 16 cases, G-band karyotyping was performed due to parents having balanced chromosomal rearrangements.
- In this study, the aCGH was performed by Aglient 8x60k microarray and only male reference DNA. Why not female reference DNA included? If gender mismatch for female sample in only male reference DNA in Agilent platforms, the microdeletion or microduplication in the sex chromosome could be missed in the software or even manual analyses data.
At the initial stages of the laboratory analysis, each fetal DNA sample was treated as a sample of unknown sex and co-hybridized with male control DNA (control DNA provided by the microarray manufacturer). If the and gender mismatch of the fetus was found in bioinformatics analysis in relation to the male reference DNA (analysis performed by CytoGenomics Software provided by Agilent), this sample was re-analyzed in silico with the female reference DNA sample.
The used procedure requires to establish of a control field on the microarray included female and male reference DNA. This approach made possible to avoid missing the microdeletions and microduplications in the sex chromosomes in case of different sex of fetal DNA.
- The parental study was mentioned in the line 105-106. Did it also go for the fetus with benign CNV?
In the Method section we explain:
Benign and likely benign CNVs were not considered for parental testing.
- In the part of results, the data analyses with many kinds of percentages in different groups were applied for results presentation, but the detailed calculations for these percentages in relevant groups were shown incompletely in the text or table or figure, even no clear statement for the same data with two different percentages, such as: two different percentages for the same CNVs data in the table 2 and in line 214-215, but no calculations for them in the table/text, and so on. The reader is easy to be confused for these results.
We have changed this section as follows:
In the group of fetuses with CVAs, chromosomal aberrations were found in 50.5% (101/202) of cases and were more frequent than in the group without CVAs (p=<0.0001) (Tab.2). Numerical chromosomal aberrations accounted for 38.1% (77/202), of which the most common was trisomy 21 (18.8%), followed by trisomy 18 (9.9%). In one case, trisomy 18 coexisted with trisomy 13, and in one case, there was a translocation (46,XY,t(6;14)(p25.3;q11.2)). In the remaining 124 fetuses, the number of chromosomes was normal, but in 27 cases (21.7%; 27/124), aCGH revealed CNVs, of which 18 (14.5%; 18/124) were classified as pathogenic CNVs, 6 (4.8%;6/124) as VOUS CNVs, and 3 (2.4%;3/124) as benign CNVs. The size of CNVs varied between 0.1Mb and 56.7Mb. The 22q11.2 microdeletion was the most common, as it occurred in 4.8% of euploid fetuses (6/124), encompassing 33.3% (6/18) of all pathogenic CNVs detected. A detailed list of CNVs and associated ultrasound findings is presented in Table 3. Pathogenic CNVs were more frequent in the CVAs group than in the group without CVAs (p=0.0003) (Tab.2). There was no significant difference between the group with CVAs and without CVAs regarding VOUS and benign CNVs.
Pathogenic CNVs were more frequent detected when there were associated ECMs (Fig.1; Tab.3). In isolated CHDs, chromosomal aberrations occurred in 17.6% (9/51) of cases, of which 5.8% were pathogenic CNVs. When CHDs were associated with vascular and other ECMs, chromosomal aberrations were observed in 76.0% (89/117) and in 11.1% (13/117) of cases that were pathogenic CNVs (Fig. 1).
- Some other issues are in the part of results.
- a) The title of figure 1 was “Cardiovascular anomalies and associated genetic disorders”.However, benign CNV is not genetic disorders.
We have changed the title of Figure to: “Cardiovascular anomalies and associated genetic findings”.
- b) In table 1, why the data for “Aortic arch anomalies” part and last part were shown differently from other parts?
We believe that the data regarding percentage of isolated aortic arch defects (without intracardiac anomalies), which can coexist with non-cardiac and / or genetic defects may help with a genetic consultation on whether to do invasive testing or not.
- c) In table 2, the case number for “Any autosomal or sex-chromosome Abnormality” was 77. However, if adding the case numbers from “Trisomy 21” to “Triploidy”, the number was 78. Furth more, there was one case with translocation, which was mentioned in line 322-323 in the discussion. But this translocation case was not shown in table 2 or the text of result.
RE: We added information about translocation in the Table 2. However still the number of all numerical anomalies is different than case number because in one case trisomy 18 and 13 coexisted.
This explanation is below the Table 2 : * Trisomy 18 and 13 coexisted in one case
- Prevalence is the rate of all cases (new and pre-existing cases) number of a disease in a specific population at a particular timepoint or over a specified period of time. Incidence is the rate of new cases of a disease occurring in a specific population over a particular period of time. In this study, it should be better to use the terms of diagnostic yield/detection rate rather than prevalence/incidence. Also, some of medical genetic terms were inconsistent or incorrect in this manuscript. For example, “genetic disorders”, “genetic anomalies”, “genetic defects” and “genetic abnormalities” all came out in this manuscript as well as “numerical chromosomal aberrations/ numerical chromosomal anomalies/ numerical chromosomal abnormalities”. In line 95, “maternal age” should be “advanced maternal age”. So, please refer to "Thompson & Thompson Genetics in Medicine" for the terminology.
Re:
We have changed incidence to diagnostic yield as well as we used chromosomal aberrations instead of “genetic disorders”, “genetic anomalies”, “genetic defects” and “genetic abnormalities”
- Please edit English language and style in this manuscript.
The manuscript has been submitted for English editing.
Reviewer 2 Report
A very detailed comprehensive analysis on an important topic. Very well presented data and suggestive tables.
These are my comments: strengths: a healthy priority topic less investigated prenatally from genetic side; very well described background and importance of CVA; good number of cases studied 1005; good highlighting of the importance of ultrasound screening findings in selecting the cases who need further genetic investigation; the 47% detection rate of a genetic background responsible by fetal CVA is suggesting the importance of doing the genetic screening during pregnancy, with increased attention for the extra cardiac symptoms and signs that can lead to identification of complex chromosomal rearrangements weakness: the recent trend is to go into WES +CNVs, obviously consider the counseling for the secondary findings ; this was slightly mentioned in the discussion; no data about NIPT available for testing these pregnancies ( especially those 38% referred for invasive procedures because of advanced maternal age) that were later identified by array as having a fetus affected by a chromosomal abnormality.
Author Response
We are grateful for your insightful comments on our paper. We have been able to incorporate changes to reflect most of the suggestions provided. We have highlighted the changes within the manuscript. Here is a point-by-point response to the reviewer' comments and concerns.
Thank you for your positive feedback and appreciation for our work put into the preparation of the manuscript.
Regarding NIPTy we added information in the Mtehod section as follows:
Each patient was also informed about the possibility of non-invasive prenatal genetic testing (NIPT) as a screening method but that these tests are not reimbursable by the Polish National Health Fund and were performed in only 21 cases. Second trimester screening was based on ultrasound only.
The manuscript has been submitted for English editing.
Reviewer 3 Report
this is a valuable piece of information, and i only wanted to add some specific suggestions.
Was genetic testing the default setting, or ‘what’ do the 1005 fetuses represent ?
As there is still word count left in the abstract, I suggest to add some information on the extent of the overall activities (otherwise, how to understand the one fifth of counseled patients).
Introduction, end of first alinea
Why not add a sentence on the non-cardiovascular associated malformations ?
Perhaps useful to add that prenatal screening does not detect all critical congenital heart diseases (about 65 % detected), so that postnatal screening on CCHD is still a useful strategy and has been implemented in a lot of countries as part of neonatal screening ? (cfr quite bold statement of the impact of TOP and IUFD, as there is still a relevant minority missed).
Methods
Some more information on the overall activity of the center are warranted.
1005 fetuses were ‘tested’, but how many were referred, but without subsequent genetic testing (or is this procedure done by default in all cases with malformations ?), as referral is commonly for diagnostic advice, not ‘only’ for genetic testing ?
What was the Down screening based on (ultrasound, NIP, triple test) ?
The information on VOUS and adult onset issues is relevant, but how were these data handled: were parents informed, or not ?
Discussion
First sentence of the second alinea, I agree, if restricted to cases undergoing invasive testing. So perhaps add this to the sentence
Author Response
We are grateful for your insightful comments on our paper. We have been able to incorporate changes to reflect most of the suggestions provided. We have highlighted the changes within the manuscript. Here is a point-by-point response to the reviewer' comments and concerns.
Was genetic testing the default setting, or ‘what’ do the 1005 fetuses represent ?
Re: we added Figure 1 presenting diagnostic flow-chart of the cases
As there is still word count left in the abstract, I suggest to add some information on the extent of the overall activities (otherwise, how to understand the one fifth of counseled patients).
We improved Abstract and clarified the results
Introduction, end of first alinea
Why not add a sentence on the non-cardiovascular associated malformations ?
Re: we added:
Previous studies have shown that extracardiac malformations (ECMs) are present in 20–60% of live-born fetuses with CHDs [15,16]
Perhaps useful to add that prenatal screening does not detect all critical congenital heart diseases (about 65 % detected), so that postnatal screening on CCHD is still a useful strategy and has been implemented in a lot of countries as part of neonatal screening ? (cfr quite bold statement of the impact of TOP and IUFD, as there is still a relevant minority missed).
Re: We agree with the above statement and in the Discussion we added:
Although most cardiac defects can be diagnosed prenatally by fetal echocardiography, many anomalies are not identified on routine prenatal ultrasound, as it was reported in the United States that only ≈30% of fetuses with serious CHDs were identified prenatally [53,54]. In turn, in Sweden, 56% of all fetuses with a critical congenital heart defect were diagnosed prenatally[55].
Methods
Some more information on the overall activity of the center are warranted.
RE: we added more information as follows:
This was a retrospective cohort study carried out at one tertiary Polish center for the prenatal diagnosis and management of fetal and neonatal pathology—the Department of Obstetrics and Perinatology in Cracow, Poland. Our center has been conducting screening for aneuploidy, established by the Polish National Health Fund, for patients from southeastern Poland since 2004. This study is a retrospective analysis of the medical records of pregnant women who had undergone invasive prenatal testing between 2018 and 2021. In the analyzed period, 12,776 pregnant women were screened, of whom 1005 were subjected after genetic counseling to invasive prenatal testing—chorionic villous sampling (15 cases) or amniocentesis (990 cases).
1005 fetuses were ‘tested’, but how many were referred, but without subsequent genetic testing (or is this procedure done by default in all cases with malformations ?), as referral is commonly for diagnostic advice, not ‘only’ for genetic testing ?
Re: we precise that during study period there were 12776 women referred to aneuploidy screening and we included only cases, which undergone invasive prenatal testing. Of course, over 1,005 women were referred for invasive procedure, but not all of them decided to do so. Unfortunately, we do not know the percentage of women who have not opted for invasive diagnostics.
What was the Down screening based on (ultrasound, NIP, triple test) ?
Re: we added:
Ultrasound examination was performed in each case before and after invasive testing. First-trimester screening and second-trimester screening were performed following Polish and international guidelines [33,34]. Nuchal translucency (NT) above the 95th percentile was considered abnormal. Additionally, the ultrasound assessment included blood flow through the tricuspid valve (TV), DV, presence/absence of nasal bone (NB), and anatomy of the fetuses. During first-trimester screening, in each case, serum markers such as free β-human chorionic gonadotrophin (free β-hCG) and pregnancy associated plasma protein A (PAPP-A) were measured according to the guidelines [33,34]. Each patient was also informed about the possibility of non-invasive prenatal genetic testing (NIPT) as a screening method but that these tests are not reimbursable by the Polish National Health Fund and were performed in only 21 cases. Second trimester screening was based on ultrasound only.
Since 2018 in our Department in cooperation with Department of Medical Genetics, Jagiellonian University Medical College aCGH is offered as a first line test for prenatal testing and in cases with ultrasound detected anomalies. aCGH with G-band karyotyping was offered in 330 cases (n=330) . Additionally, in 16 cases G-band karyotyping was performed due to parents with balanced chromosomal rearrangements.
The information on VOUS and adult onset issues is relevant, but how were these data handled: were parents informed, or not ?
Re: Yes, parents were informed and counselled. In Method section we explained:
Benign and likely benign cases were not reported in our results. In addition, in cases with fetal pathogenic and VOUS CNVs, parents were counseled to perform aCGH to verify if the fetal CNVs were inherited or appeared de novo. Benign and likely benign CNVs were not considered for parental testing.
Discussion
First sentence of the second alinea, I agree, if restricted to cases undergoing invasive testing. So perhaps add this to the sentence
RE: Yes, we agree and added as follows:
In this retrospective cohort study of more than 1000 fetuses undergoing invasive prenatal testing, we look in detail at the diagnostic yield of cardiovascular anomalies, associated non-cardiac structural defects, and chromosomal aberrations.
Round 2
Reviewer 1 Report
Thanks for revising the manuscript and the response to the review comments. However, the response and the revised manuscript have failed to address the main flaws in this study.